# Structural Phase Diagram of LaO$_{1-x}$F$_x$BiSSe: Suppression of the Structural Phase Transition by Partial F Substitutions

**Kazuhisa Hoshi [1], Shunsuke Sakuragi [2], Takeshi Yajima [2], Yosuke Goto [1], Akira Miura [3], Chikako Moriyoshi [4], Yoshihiro Kuroiwa [4] and Yoshikazu Mizuguchi [1,*]**

[1]   Department of Physics, Tokyo Metropolitan University, Hachioji 192-0379, Japan; hoshi-kazuhisa@ed.tmu.ac.jp (K.H.); y_goto@tmu.ac.jp (Y.G.)
[2]   Institute for Solid State Physics, University of Tokyo, Kashiwa 277-8581, Japan; sakuragi@issp.u-tokyo.ac.jp (S.S.); yajima@issp.u-tokyo.ac.jp (T.Y.)
[3]   Faculty of Engineering, Hokkaido University, Sapporo 060-8628, Japan; amiura@eng.hokudai.ac.jp
[4]   Graduate School of Advanced Science and Engineering, Hiroshima University, Higashihiroshima 739-8526, Japan; moriyosi@sci.hiroshima-u.ac.jp (C.M.); kuroiwa@sci.hiroshima-u.ac.jp (Y.K.)
*   Correspondence: mizugu@tmu.ac.jp

**Abstract:** Recently, the anomalous two-fold-symmetric in-plane anisotropy of superconducting states has been observed in a layered superconductor system, LaO$_{1-x}$F$_x$BiSSe ($x = 0.1$ and $0.5$), with a tetragonal (four-fold symmetric) in-plane structure. To understand the origin of the phenomena observed in LaO$_{1-x}$F$_x$BiSSe, clarification of the low-temperature structural phase diagram is needed. In this study, we have investigated the low-temperature crystal structure of LaO$_{1-x}$F$_x$BiSSe ($x = 0, 0.01, 0.02, 0.03$, and $0.5$). From synchrotron X-ray diffraction experiments, a structural transition from tetragonal to monoclinic was observed for $x = 0$ and $0.01$ at 340 and 240 K, respectively. For $x = 0.03$, a structural transition and broadening of the diffraction peak were not observed down to 100 K. These facts suggest that the structural transition could be suppressed by 3% F substitution in LaO$_{1-x}$F$_x$BiSSe. Furthermore, the crystal structure for $x = 0.5$ at 4 K was examined by low-temperature laboratory X-ray diffraction, which confirmed that the tetragonal structure is maintained at 4 K for $x = 0.5$. Our structural investigation suggests that the two-fold-symmetric in-plane anisotropy of superconducting states observed in LaO$_{1-x}$F$_x$BiSSe was not originated from structural symmetry lowering in its average structure. To evaluate the possibility of the local structural modification like nanoscale puddles in the average tetragonal structure, further experiments are desired.

**Keywords:** BiCh$_2$-based superconductor; layered superconductor; crystal structure; phase transition; structural phase diagram

## 1. Introduction

BiCh$_2$-based (Ch = S, Se) superconductors were discovered in 2012 [1,2], which was followed by the development of related layered compounds (material developments are summarized in a recent review article [3]). The crystal structure of typical BiCh$_2$-based superconductor is composed of an alternate stacking of an insulating layer and a BiCh$_2$ conducting bilayer, which is similar to those of high-temperature superconductors such as cuprates and iron-based superconductors (IBSCs) [4,5]. In particular, REOBiCh$_2$-type (RE = Rare earth elements) compounds have been extensively studied owing to its flexibility on elemental substitution of constituent elements. On the electronic characteristics, a parent phase of REOBiCh$_2$ is a semiconductor with a band

gap. Electron carrier doping by a partial substitution of F for the O site has been used to induce metallicity and then superconductivity in the system [2,3]. The pairing mechanisms of the superconductivity in the $BiCh_2$-based systems are still controversial; both conventional and unconventional mechanisms have been proposed from both theoretical and experimental studies [6]. For example, investigations on thermal conductivity, specific heat, and magnetic penetration depth have suggested a conventional model of superconductivity for $BiCh_2$-based superconductors in the early stage [7–9]. However, an angle-resolved photoemission spectroscopy (ARPES) study reported the observation of the anisotropic superconducting gap, indicating that unconventional superconductivity is emerging in $NdO_{0.71}F_{0.29}BiS_2$ [10]. Furthermore, the absence of isotope effects was observed in tetragonal $LaO_{0.6}F_{0.4}Bi(S,Se)_2$, which was examined using $^{76}Se$ and $^{80}Se$ isotopes [11], and $Bi_4O_4S_3$, which was examined using $^{32}S$ and $^{34}S$ isotopes [12]. These results also imply that unconventional pairing is essential for $BiCh_2$-based superconductors with a tetragonal structure. Notably, the conventional-type isotope effect was observed in the monoclinic (high-pressure) phase of $(Sr,La)FBiS_2$ [13]. Therefore, structural symmetry may be a switch of pairing symmetry of superconductivity in $BiCh_2$-based compounds, and hence further investigation on the relationship between the crystal structure (local structure) and mechanisms of superconductivity should deeply be studied for the $BiCh_2$-based superconductor family.

Recently, two-fold symmetry in the *ab*-plane (in-plane) anisotropy of the magnetoresistance (MR) was observed in superconducting states of $BiCh_2$-based $LaO_{1-x}F_xBiSSe$ ($x$ = 0.1 and 0.5) [14,15]. Through room-temperature structural analysis using X-ray diffraction (XRD), the crystal structure was determined to be a tetragonal type (*P4/nmm*) having four-fold symmetry in the *ab*-plane [16]. Therefore, the two-fold-symmetric in-plane anisotropy of MR is expected to be breaking its structural symmetry in the conducting plane. This phenomenon is quite similar to what was observed in *nematic superconductors*, $A_xBi_2Se_3$ (A = Cu, Sr, Nb) and IBSCs [17–22]. In those nematic superconductors, the two-fold-symmetric anisotropy of physical properties, including MR, magnetization, specific heat, and superconducting gap, in superconducting states has also been observed, in spite of three-fold ($A_xBi_2Se_3$) or four-fold (IBSCs) symmetry in its crystal structure. Since in-plane symmetry breaking in superconducting properties can be related to the possible structural symmetry lowering, the determination of low-temperature crystal structure is needed to conclude the origin of the two-fold-symmetric superconducting properties in layered superconductors. For $LaO_{1-x}F_xBiSSe$, however, there has still been a possibility of the emergence of the two-fold-symmetric in-plane anisotropy of MR due to structural symmetry lowering because the parent phase (F-free) LaOBiSSe undergoes a structural transition from tetragonal (high-*T* phase: *P4/nmm*) to monoclinic (low-*T* phase: *P2$_1$/m*) at 300–400 K (see Figure 1e) [3,16]. Furthermore, $BiS_2$-based $LaO_{0.5}F_{0.5}BiS_2$ shows a pressure-induced structural transition from tetragonal to monoclinic [23]. Since the origin of those structural transitions can be linked to the activity of the Bi lone-pair electrons, which affect local structures of the conducting $BiCh_2$ layers [24], the structural instability could be present for all $BiCh_2$-based systems [25,26]. It has been known that carrier doping via partial element substitution suppresses the structural transition and stabilizes tetragonal structure in a $REOBiCh_2$-type structure [3]. Additionally, theoretical analysis on the stability of the crystal structure as a function of carrier concentration suggested that the tetragonal structure is more stabilized than the monoclinic one in electron-doped (F-substituted) $LaOBiS_2$ [27].

On the basis of those facts, systematic analyses on the crystal structure near the cross-over between tetragonal and monoclinic structures of $LaO_{1-x}F_xBiSSe$ are, therefore, needed to classify the two-fold-symmetric in-plane anisotropy of MR in the superconducting states as nematic superconductivity. Here, we have studied the temperature and carrier concentration dependences of crystal structure of $LaO_{1-x}F_xBiSSe$. A structural transition from tetragonal to monoclinic was observed at 340 K for $x$ = 0 and at 240 K for $x$ = 0.01. No structural transition and X-ray diffraction peak broadening were observed for $x$ = 0.03 down to 100 K. These results suggest that the structural transition is rapidly suppressed by carrier doping and disappears at $x$~0.03 in $LaO_{1-x}F_xBiSSe$. Furthermore, we have confirmed that the tetragonal structure has been maintained at 4 K for $LaO_{1-x}F_xBiSSe$ ($x$ = 0.5).

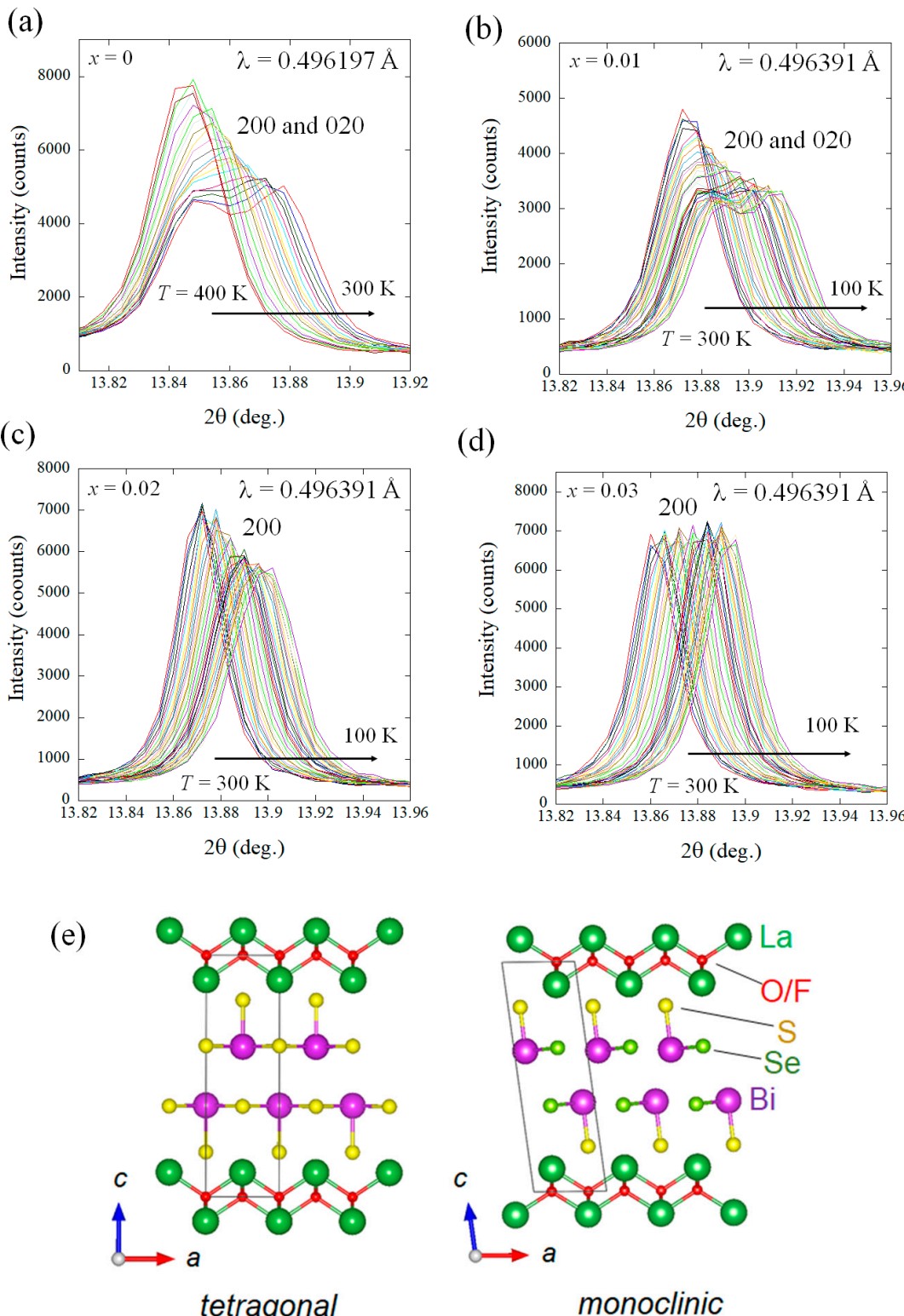

**Figure 1.** Temperature evolutions of the (200) and (020) peaks of the synchrotron X-ray diffraction (SXRD) patterns for (**a**) $x = 0$ (**b**) $x = 0.01$, (**c**) $x = 0.02$, and (**d**) $x = 0.03$ of LaO$_{1-x}$F$_x$BiSSe. The wavelengths used in the scanning are indicated in the figures. (**e**) Schematic images of structural difference between tetragonal and monoclinic phases of LaO$_{1-x}$F$_x$BiSSe.

## 2. Results

In a $REOBiCh_2$-type structure, electron carrier doping results in the compression of the *c*-axis [3]. The lattice constant *c* for the examined $LaO_{1-x}F_xBiSSe$ ($x$ = 0, 0.01, 0.02, and 0.03) shows a systematic decrease with increasing nominal $x$, as shown in Figure S4 (supporting materials), which suggests that electron carriers are systematically doped in the low F-doping regime.

As reported in [3,16], the parent phase LaOBiSSe ($x$ = 0) undergoes a structural transition from tetragonal ($P4/nmm$) to monoclinic ($P2_1/m$) at 300–400 K. In a monoclinic phase, the (200) peak splits into (200) and (020). Therefore, by scanning the (200) peak on temperature, the structural transition temperature ($T_s$) and the evolution of in-plane lattice constants (*a* and *b*) can be investigated. Figure 1a–d display the temperature dependences of the tetragonal (200) peak and monoclinic (200) and (020) peaks on the synchrotron X-ray diffraction (SXRD) patterns for $x$ = 0, 0.01, 0.02, and 0.03, respectively. Commonly, the (200) peak shifts to higher angles on cooling, which is due to the compression of the lattice. As shown in Figure 1a, splitting of the (200) peak into the monoclinic (200) and (020) peaks was observed below 340 K for $x$ = 0, which indicates a structural transition to the monoclinic structure at $T_s$ = 340 K. Note that the peak intensity for $x$ = 0 is rapidly suppressed with decreasing temperature as the temperature approached $T_s$. This is a signature of structural instability toward a structural transition (symmetry lowering) because the decrease in peak intensity during the temperature scanning corresponds to the broadening of the peak in this experimental setup, in which the sample condition was not modified and the temperature was continuously changed. A structural transition was observed for $x$ = 0.01 at 240 K. A similar trend on the suppression of peak intensity was observed in Figure 1b. For $x$ = 0.02 and 0.03, a clear structural transition was not observed down to 100 K. However, the suppression of the peak intensity was observed for $x$ = 0.02. This signature implies that the sample has the structural instability, and a structural transition is expected below 100 K. Notably, the peak intensity is almost constant from 300 to 100 K for $x$ = 0.03, which implies that no structural transition is expected at temperatures lower than 100 K. To check the evolution of the peak broadening, the temperature evolutions of the full width half maximum (FWHM) of the (200) peaks estimated from the Gaussian fitting are also consistent with the scenario above (see Figure S3 of supporting materials). These results suggest that the structural transition can be completely suppressed at concentration lower than $x$ = 0.03 in $LaO_{1-x}F_xBiSSe$.

To analyze lattice constants *a* and *b* from the data shown in Figure 1, the (200) and (020) peaks were fitted by one or two Gaussian functions. Two Gaussian functions were used for $x$ = 0 and 0.01, where a clear structural transition was observed. For $x$ = 0.02 and 0.03, we analyzed the lattice constant with one Gaussian function. Figure 2a shows the temperature dependence of the lattice constants *a* and *b* for $x$ = 0, which clearly shows a transition at $T_s$ = 340 K. As shown in Figure 2b, the $T_s$ for $x$ = 0.01 was 240 K. For $x$ = 0.02 and 0.03, the lattice constant *a* linearly changed with decreasing temperature, which implies that the tetragonal structure is dominant in this temperature regime. The trend that the structural transition from tetragonal to monoclinic is rapidly suppressed by F substitution in $LaO_{1-x}F_xBiSSe$ is consistent with the theoretical study which proposed that the tetragonal structure is more stable than monoclinic in F-substituted $LaOBiS_2$ [27].

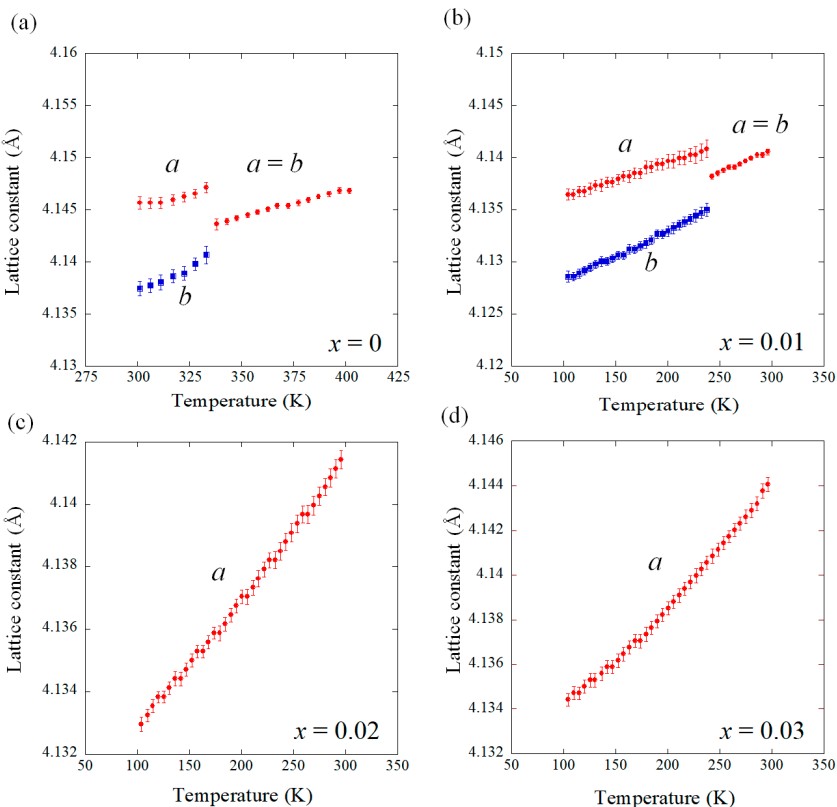

**Figure 2.** Temperature dependences of the lattice constants *a* and *b* for LaO$_{1-x}$F$_x$BiSSe: (**a**) $x = 0$, (**b**) $x = 0.01$, (**c**) $x = 0.02$, and (**d**) $x = 0.03$.

Figure 3 shows a structural phase diagram of LaO$_{1-x}$F$_x$BiSSe. Due to the experimental limitation, we could scan the lattice constant on temperature at $T > 100$ K only. From the evidence of the suppression of peak intensity, a structural transition below 100 K was assumed for $x = 0.02$. In contrast, because the peak intensity for $x = 0.03$ does not show a decrease down to 100 K, we assumed that the low-temperature structure for $x = 0.03$ is tetragonal down to 0 K, which is indicated with a cross symbol in Figure 3.

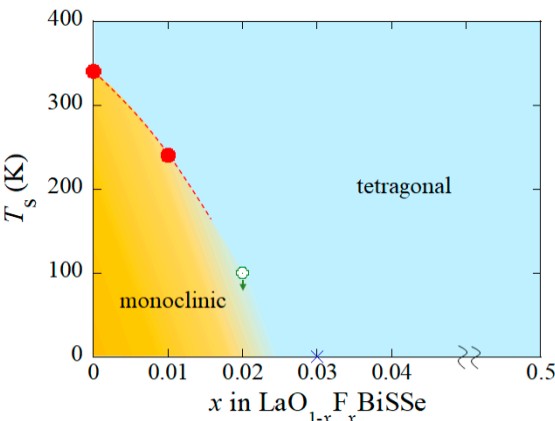

**Figure 3.** Structural phase diagram of LaO$_{1-x}$F$_x$BiSSe. A circle symbol with an arrow at $x = 0.02$ indicates that the $T_s$ for $x = 0.02$ is lower than 100 K. A cross symbol at $x = 0.03$ has been plotted under the assumption that the sample with $x = 0.03$ does not undergo a structural transition down to 0 K.

To investigate the influence of the structural transition on the transport properties, the temperature dependence of electrical resistivity was measured for $x = 0$ ($T_s = 340$ K), 0.01 ($T_s = 240$ K), 0.02, and 0.03

(no transition is expected) and plotted in Figure 4. As reported in [16], an upturn, non-metallic behavior is observed below 90 K for $x = 0$. It was reported that no anomaly was observed at 340 K in the high-temperature resistivity measurements [28]. For $x = 0.01$, metallic-like behavior was observed while a small upturn is observed at a low temperature. Notably, there is no clear anomaly at $T_s$, as indicated by an arrow, where a structural transition was detected (see Figures 1 and 2). The absence of anomaly in the resistivity data is probably because of the small distortion of the in-plane structure ($a/b$ ratio in the low-$T$ phase) below the $T_s$. The $a/b$ ratio for $x = 0.01$ is 1.002, which indicates 0.2% in-plane distortion. For example, this is clearly smaller than 0.8% in-plane distortion that observed in layered compound $BaFe_2As_2$ [29], in which a clear anomaly is observed in its resistivity–temperature curve. For $x = 0.02$ and 0.03, a typical metallic behavior is observed. In addition, a small hump is observed at $T = 150–200$ K for $x = 0.03$. Since no structural anomaly was observed for $x = 0.03$ in the temperature range, we have no explanation about the anomaly at present. However, a similar anomaly has been reported for several Bi-chalcogenide layered compounds [30–32], and possible charge–density–wave (CDW) ordering has been suggested for $EuFBiS_2$ [33].

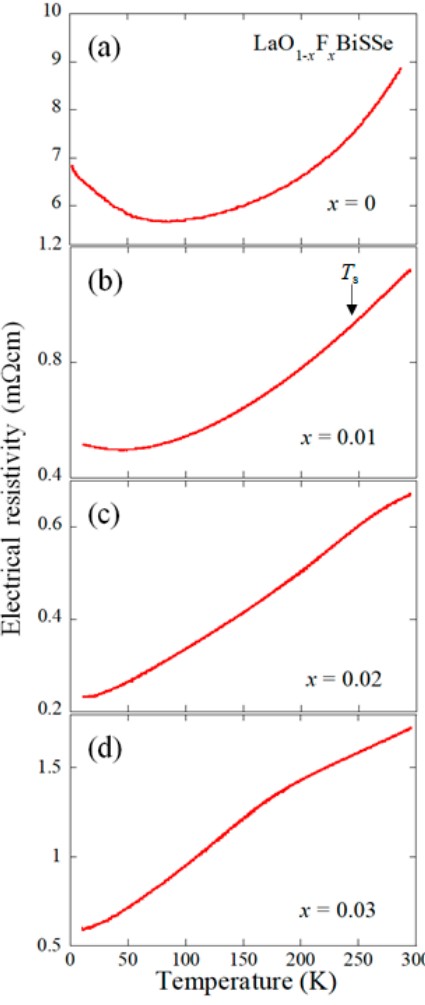

**Figure 4.** Temperature dependences of resistivity for (**a**) $x = 0$, (**b**) $x = 0.01$, (**c**) $x = 0.02$, and (**d**) $x = 0.03$ of $LaO_{1-x}F_xBiSSe$. $T_s$ denotes the structural transition temperature for $x = 0.01$.

## 3. Discussion

From the results described above, it is reasonable to expect that the low-temperature crystal structure for $x = 0.1$ and 0.5, in which the two-fold-symmetric in-plane anisotropy of MR in the superconducting states was observed [14,15], is tetragonal with four-fold symmetry in *ab*-plane.

To confirm this assumption, low-temperature laboratory XRD experiments were performed for $x = 0.5$ at 4 K. Figure 5a shows the (200) peaks at $T = 4$ and 300 K collected with a Cu-K$\alpha$ radiation using a pelletized sample. The peak at 4 K shifts to a higher angle because of lattice compression by cooling. Neither peak splitting nor broadening was observed for the (200) peak, indicating that the tetragonal structure is maintained at 4 K for $x = 0.5$. Figure 5b shows the 004 peaks collected at $T = 4$ and 300 K for $x = 0.5$. No peak broadening is observed for the (004) peak at 4 K. From the structural investigations for LaO$_{1-x}$F$_x$BiSSe shown here, we suggest that the origin of the two-fold-symmetric in-plane anisotropy of MR in the superconducting states of LaO$_{0.5}$F$_{0.5}$BiSSe is not structural symmetry lowering in its average structure.

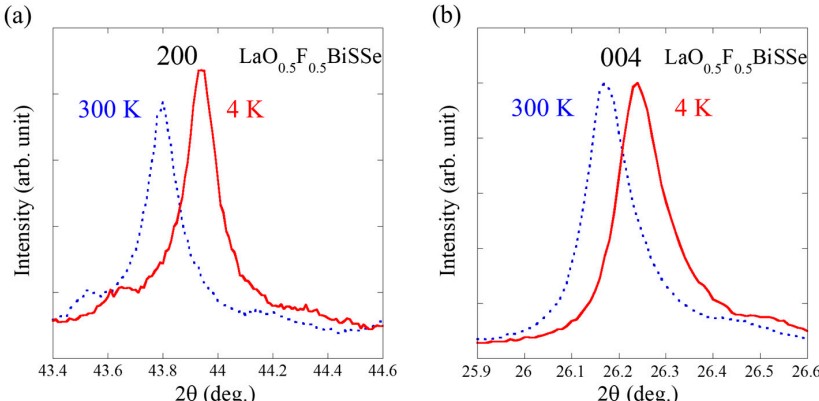

**Figure 5.** (**a**) (200) peak and (**b**) (004) peak in a laboratory XRD pattern for LaO$_{0.5}$F$_{0.5}$BiSSe ($x = 0.5$) collected at 4 and 300 K.

Herein, we briefly describe prospects for the BiCh$_2$-based systems as a new avenue to investigate nematic superconductivity. So far, the major investigations into nematic superconductivity have been focused on doped Bi$_2$Se$_3$ or IBSCs as described in the introduction. One of the commonalities in Bi$_2$Se$_3$, IBSCs and the BiCh$_2$-based systems is multi-orbital nature. As mentioned in the introduction, structural instability exists in BiCh$_2$-based compounds with a tetragonal structure. Although there have been no results indicating the importance of orbital fluctuations to superconductivity mechanisms in the system, we may reach the scenario from the analogy with IBSCs, in which orbital fluctuation plays important roles in superconductivity mechanisms [34]. If the orbital fluctuations in BiCh$_2$-based compounds are related to the emergence of nematic superconductivity, the BiCh$_2$-based system will be useful for understanding the mechanisms of the emergence of nematic superconductivity in layered superconductors with the multi-orbital nature. Another possible commonality is the relation to topological superconductivity states. In doped Bi$_2$Se$_3$ and IBSCs, topological superconducting states have been proposed [35,36]. Although there is still no experimental evidence of topological superconductivity in BiCh$_2$-based systems, a theoretical study suggested the possibility of weak topological superconductivity in BiCh$_2$-based compounds [37]. To clarify the mechanisms of superconductivity in BiCh$_2$-based systems and to find commonalities to the other nematic superconductors, further theoretical and experimental investigations are needed.

Lastly, we briefly mention the possibility of the presence of local structural modification like structural nanoscale puddles in the average tetragonal structure [38–40], which cannot be detected by XRD and can explain the two-symmetric in-plane anisotropy in layered tetragonal superconductors with interlayer structural mismatch [41]. Therefore, to evaluate the possibility of the local structural modification like nanoscale puddles in the average tetragonal structure, further experiments are desired.

## 4. Materials and Methods

Polycrystalline samples of LaO$_{1-x}$F$_x$BiSSe ($x = 0$, 0.01, 0.02, 0.03, and 0.5) were prepared using a solid-state-reaction method. Bi$_2$S$_3$ and Bi$_2$Se$_3$ were pre-synthesized through reacting Bi (99.999%),

S (99.9999%), and Se (99.999%) grains. Powders of $La_2O_3$ (99.9%), $La_2S_3$ (99.9%), $BiF_3$ (99.9%), $Bi_2S_3$, and $Bi_2Se_3$, and Bi (99.999%) grains with nominal compositions of $LaO_{1-x}F_xBiSSe$ were mixed, pressed into a pellet, sealed into an evacuated quartz tube, and annealed at 700 °C for 15 h. The obtained sample was mixed for homogenization, pressed into a pellet, sealed into an evacuated quartz tube, and annealed at 700 °C for 15 h. Schematic images of crystal structure were depicted using VESTA [42]. Synchrotron X-ray diffraction (SXRD) experiments were performed from 400 to 300 K for $x = 0$ and from 300 to 100 K under the temperature control system with nitrogen gas for $x = 0.01$, 0.02, and 0.03 at the beamline BL02B2 of SPring-8 under research proposals Nos. 2019A1114 ($\lambda = 0.496197$ Å) and 2019B1195 ($\lambda = 0.496391$ Å). For $x = 0.5$, high-resolution powder X-ray diffraction (XRD) experiments with a Cu-K$\alpha$1 radiation monochromatized by a Ge (111)-Johansson-type monochromator at 300 and 4 K were performed on a SmartLab diffractometer equipped with a GM refrigerator. The typical XRD patterns of both SXRD and conventional XRD experiments are shown in the supporting materials (Figures S1 and S2). The temperature dependence of electrical resistivity was measured using four-terminal method with a DC current of 1 mA on a GM refrigerator.

## 5. Conclusions

We have investigated low-temperature crystal structure of $BiCh_2$-based compounds $LaO_{1-x}F_xBiSSe$ ($x = 0$, 0.01, 0.02, 0.03, and 0.5). From SXRD experiments, a structural transition from tetragonal to monoclinic was observed for $x = 0$ and 0.01. For $x = 0.03$, a structural transition and broadening of the diffraction peak were not observed down to 100 K. These facts suggest that the structural transition could be suppressed by 3% F substitution in $LaO_{1-x}F_xBiSSe$. Furthermore, from XRD experiments at $T = 4$ K, the crystal structure for $x = 0.5$ at 4 K was determined as tetragonal. The structural phase diagram obtained in this study suggests that the two-fold-symmetric in-plane anisotropy of superconducting states observed in $LaO_{1-x}F_xBiSSe$ was not originated from structural symmetry lowering in its average structure. To evaluate the possibility of the local structural modification like nanoscale puddles in the average tetragonal structure, further experiments are desired.

**Supplementary Materials:** The following are available online at http://www.mdpi.com/2410-3896/5/4/81/s1, Figure S1: synchrotron XRD profiles, Figure S2: Low-temperature XRD profiles, Figure S3: Estimated FWHM of the (200) peak, Figure S4: Lattice constant *c*.

**Author Contributions:** Conceptualization, K.H. and Y.M.; methodology, K.H., S.S., T.Y., Y.G., A.M., C.M., Y.K. and Y.M.; validation, K.H. and Y.M.; formal analysis, K.H., S.S., T.Y., Y.G., A.M., C.M., Y.K. and Y.M.; investigation, K.H., S.S., T.Y., Y.G., A.M., C.M., Y.K. and Y.M.; resources, K.H., Y.G. and Y.M.; data curation, K.H. and Y.M.; writing—original draft preparation, K.H. and Y.M.; writing—review and editing, K.H., S.S., T.Y., Y.G., A.M., C.M., Y.K., Y.G. and Y.M.; visualization, K.H. and Y.M.; supervision, Y.G. and Y.M.; project administration, Y.M.; funding acquisition, Y.G. and Y.M. All authors have read and agreed to the published version of the manuscript.

**Funding:** This research was partly funded by Grants in Aid for Scientific Research (KAKENHI), grant numbers 15H05886, 16H04493, 18KK0076, and 19K15291, JST-CREST, grant number JPMJCR20Q4, and Tokyo Metropolitan Government Advanced Research, grant number H31-1. The APC was funded by Tokyo Metropolitan University.

**Acknowledgments:** The authors thank O. Miura for support in sample characterization.

**Conflicts of Interest:** The authors declare no conflict of interest. The funders had no role in the design of the study; in the collection, analyses, or interpretation of data; in the writing of the manuscript, or in the decision to publish the results.

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
