# Peer review of "Structural Phase Diagram of LaO1−xFxBiSSe: Suppression of the Structural Phase Transition by Partial F Substitutions"

_condensedmatter, doi:10.3390/condmat5040081_

Round 1
Reviewer 1 Report
the submitted manuscript ID: condensedmatter-1023018 titled "Structural phase diagram of LaO1-xFxBiSSe: suppression of the structural phase transition by partial F substitutions by Kazuhisa Hoshi, Shunsuke Sakuragi, Takeshi Yajima, Yosuke Goto, Akira Miura, Chikako Moriyoshi, Yoshihiro Kuroiwa, Yoshikazu Mizuguchi report a X-ray diffraction (XRD) experiments using synchrotron radiation and in-house X-ray sources of low-temperature crystal structure of LaO1-xFxBiSSe (x = 0, 0.01, 0.02, 0.03, and 0.5). For the x=0.03 and x = 0.5 the powder XRD data do not show the structural phase transition from tetragonal to monoclinic.The experimental results are if high interest today, therefore the paper could be published in Condensed Matter with major changes.
The conclusions and the abstract need to be improved since they show a prejudice which is not justified by the experiment in favor of the theory of nematicity aganist the theory of pairing driven by Fano resonances near Lifshitz transitions in complex quantum matter. In fact, the presence of structural nanoscale puddles in the average tetragonal structure detected by standard XRD could explain two-fold-symmetric in-plane anisotropy of the magnetoresistance and in the k-dependent anisotropic superconducting gap in short coherence length superconductors.
Tajima, H., Yerin, Y., Perali, A., & Pieri, P. (2019). Enhanced critical temperature, pairing fluctuation effects, and BCS-BEC crossover in a two-band Fermi gas. Physical Review B, 99(18), 180503.
Salasnich, L., Shanenko, A. A., Vagov, A., Aguiar, J. A., & Perali, A. (2019). Screening of pair fluctuations in superconductors with coupled shallow and deep bands: A route to higher-temperature superconductivity. Physical Review B, 100(6), 064510.
Mazziotti, M. V., Valletta, A., Campi, G., Innocenti, D., Perali, A., & Bianconi, A. (2017). Possible Fano resonance for high-Tc multi-gap superconductivity in p-Terphenyl doped by K at the Lifshitz transition. EPL (Europhysics Letters), 118(3), 37003.
Therefore, the authors are invited to improve their abstract and conclusions by reminding the reader that "nanoscale puddles with two-fold-symmetric in-plane anisotropy“ are not detected using their experimental method but further work need to be done using different experimental methods looking for nanoscale puddles.
in fact nanoscale puddles have been observed by neutrons in tetragonal phase of La2− x (Sr, Ba) xCuO4
Wakimoto, S., Kimura, H., Fujita, M., Yamada, K., Noda, Y., Shirane, G., ... & Birgeneau, R. J. (2006). Incommensurate lattice distortion in the high temperature tetragonal phase of La2− x (Sr, Ba) xCuO4. Journal of the Physical Society of Japan, 75(7), 074714-074714.
in tetragonal HgBa2CuO4 + y by scanning XRD
Campi, G., Bianconi, A., Poccia, N., Bianconi, G., Barba, L., Arrighetti, G., ... & Burghammer, M. (2015). Inhomogeneity of charge-density-wave order and quenched disorder in a high-T c superconductor. Nature, 525(7569), 359-362.
and in tetragonal Y123
Campi, G., Ricci, A., Poccia, N., Barba, L., Arrighetti, G., Burghammer, M., ... & Bianconi, A. (2013). Scanning micro-x-ray diffraction unveils the distribution of oxygen chain nanoscale puddles in YBa 2 Cu 3 O 6.33. Physical Review B, 87(1), 014517.
In fact, like in perovkites the presence of a lattice mismatch
Agrestini, S., Saini, N. L., Bianconi, G., & Bianconi, A. (2003). The strain of CuO2 lattice: the second variable for the phase diagram of cuprate perovskites. Journal of Physics A: Mathematical and General, 36(35), 9133.
between the insulating layer and the BiCh2 conducting bilayer in BiCh2 -based superconductor tuned by doping could drive the transition to the tetragonal phase but nanoscale anisotropic puddles with correlated disorder are expected to be present in the tetragonal phase.
Finally, the authors could consider to change their sentence in the conclusions
1) "the two-fold-symmetric in-plane anisotropy of superconducting states observed in LaO1-xFxBiSSe was not originated from structural symmetry lowering"
2) "hence nematic superconductivity is emerging in the system"
by mentioning that other authors have proposed an alternate theory of "multigaps pairing driven by Fano resonances" in the possible presence of nanoscale puddles which could be detected by different experimental methods.
Author Response
Thank you so much for reviewing our manuscript. Please see the attached.

Reviewer 2 Report
Please see attached file

Author Response

(The authors gave the same response as above.)

Reviewer 3 Report
The article condensed matter-1023018, “Structural phase diagram of LaO1-xFxBiSSe: suppression of the structural phase transition by partial F substitution”, reports the low –T synchrotron X-ray for x = 0.01, 0.02, and 0.03, getting the result that the structure transition could be suppressed by 3%-doping. They also rule out the possibility of nematic behavior originating from the symmetry lowering. The result is important and informative, however, several parts need to be clearer.
My comments list below:
- The system change of the X-ray patterns of these doping samples is needed to show that O is effectively substituted by F. The doping level is very low, just 1 - 3 percent, how to control and clarify it needs to be discussed.
- The authors do not have X-ray diffraction data lower than 100 K, especially for sample 0.02 and 0.03. The conclusion of the cross of the phase transition between 0.02 and 0.03 is just an assumption based on that no suppression of the peak intensity of the 0.03 sample is observed. As the low T data at 4 K for x=0.5 sample was shown, the conventional or synchrotron low-T XRD data should also be presented as helpful evidence.
- The Ts for sample x= 0 and 0.01 are 340 K and 240 K, respectively. Did the authors see any signal in the temperature dependence of resistivity? The resistivity data above 300 K needs to be added at least for the parent sample.
- A small hump is shown for sample 0.03 in the article at 150 – 200 K. Did the authors notice that there is a similar feature around 250 K for sample x=0.02? The c/a ratio was reported to have a connection with this CDW-like behavior in ref. 32. Is there any connection in this system?
Minor points
- It is better to change 200 peaks to (200) peaks to avoid misunderstanding.
- Line 87 “were not observed” should be “were observed”.
- Please check the manuscript carefully and correct other clerical and grammatical errors.

Author Response

(The authors gave the same response as above.)

Round 2
Reviewer 2 Report
Please accept the manuscript in its present form.
Reviewer 3 Report
All the questions are fully addressed.
I recommend accepting this paper in present form.